# The development, feasibility and credibility of intra-abdominal pressure measurement techniques: A scoping review

ZhiRu Li[1], HuaFen Wang[1]*, FangYan Lu[2]

1 Nursing Department, The First Affiliated Hospital, Zhejiang University School of Medicine, Hangzhou, Zhejiang, China, 2 Hepatobiliary and pancreatic surgery, the First Affiliated Hospital, Zhejiang University School of Medicine, Hangzhou, Zhejiang, China

* 2185015@zju.edu.cn

## Abstract

### Aim

To provide a comprehensive overview on emerging direct and alternative methods for intra-abdominal pressure (IAP) measurement techniques.

### Methods

This was a scoping review study following Arksey and Malley's framework. The PubMed, EMBASE, Web of Science, EBSCO, Scopus and ProQuest databases were searched, and we only considered studies published from 2000 as we have extended the data from two previous reviews. Original studies that reported on the development, feasibility and credibility of IAP measurement techniques were included.

### Results

Forty-two of 9954 screened articles were included. IAP measurement techniques include three major categories: direct, indirect and less invasive measurement techniques. Agreement analyses were performed in most studies, and some explored the safety, time expenditure and reproducibility of IAP measurement techniques.

### Conclusions

Clinical data assessing the validation of new IAP measurement techniques or the reliability of established measurement techniques remain lacking. Considering the cost and invasiveness, direct measurement is not recommended as a routine method for IAP measurement and should be preserved for critically ill patients where standard techniques are contraindicated or could be inaccurate. The measurement accuracy, reliability and sensitivity of the transrectal and transfemoral vein methods remain insufficient and cannot be recommended as surrogate IAP measures. Transvesical measurement is the most widely used method, which is the potentially most easy applicable technique and can be used as a reliable method for continuous and intermittent IAP measurement. Wireless transvaginal method

**Data Availability Statement:** The minimal data set underlying the results described in this paper can be found with DOI: 10.5061/dryad.z34tmpgmv.

**Funding:** The author(s) received no specific funding for this work.

**Competing interests:** The authors have declared that no competing interests exist.

facilitates the quantitative IAP measurement during exercise and activity, which laying the foundations for monitoring IAP outside of the clinic environment, but the accuracy of this technique in measuring absolute IAP cannot be determined at present. Less invasive technology will become a new trend to measure IAP and has substantial potential to replace traditional IAP measurement technologies, but further validation and standardization are still needed. Medical professionals should choose appropriate measurement tools based on the advantages and disadvantages of each IAP technique in combination with assessing specific clinical situations.

## Introduction

Intra-abdominal pressure (IAP) is the steady-state pressure in the abdominal cavity caused by the interaction between the abdominal wall and internal organs [1].One study showed that within one week of admission, approximately 50% of critically ill will patients develop intra-abdominal hypertension (IAH), which is defined as a sustained or repeated pathological elevation in IAP>12 mmHg (>10 mmHg in children) [2–4]. IAH has harmful consequences on local and systemic tissues and organ systems, such as impaired ventilation, acute renal failure, decreased blood flow to organs and haemodynamic instability, which is considered a key reason for the high morbidity and mortality rates among ICU patients [5]. The clinical situation in which at least one new or worsening organ dysfunction results from IAH is referred to as abdominal compartment syndrome (ACS) [2]. In a 1-day prevalence study of IAH involving 13 intensive care units across 6 countries, 58.8% of the patients developed IAH, and 8.2% had ACS [6].

Reliable IAP measurement is important; at the level of individual patients, the use of decompressive laparotomy is often based on absolute IAP values. Additionally, accurate IAP measurements are extremely important to further explore clinical correlation between IAH and organ function or organ failure [7]. IAP can be measured directly or indirectly via other intra-abdominal organs [8]. However, each measurement method should be assessed on the basis of its advantages and disadvantages. For example, the direct measurement method may be considered more accurate, but it is invasive and complex; The transvesical measurement serves as the "gold standard" for indirect measurement of IAP, but is susceptible to factors such as body position and bladder compliance [9]. Therefore, healthcare providers must be able to choose appropriate IAP measurement techniques based on the patient's condition, available measuring equipment, specific clinical context, and measure IAP in a standardized and reproducible fashion.

In 2006, the World Society of the Abdominal Compartment Syndrome (WSACS) recommended transvesical IAP measurement, described by Kron et al. in 1984 as the preferred method for indirect IAP measurement in critically ill patients [10]. However, some researchers have questioned this method, especially regarding dynamic measurements [6]. Although IAP measurements can be obtained at regular intervals using this technology, it remains labour intensive because it is time-consuming and observer-dependent, especially when hourly IAP measurements are required [11]. According to the IAH/ACS management algorithm recommended by WSACS, if ICU patients have the risk factors for IAH/ACS, their IAP should be monitored continuously or at least every 4–6 hours; in patients with evolving organ dysfunction, this measurement frequency should be increased to hourly [2]. Given the frequent requests for IAP monitoring in ICU patients, it is essential to identify a simple, accurate, rapid,

and less invasive IAP monitoring method. Some researches rather suggested non-invasive techniques to measure IAP [12, 13]. Numerous direct and alternative methods for IAP measurements have been developed in recent decades. However, the repeatability, safety, and credibility of these measurement techniques remain unclear. Thus, in this study, the commonly used scope review report framework proposed by Arksey and Malley [14] was adopted to comprehensively collect direct and alternative IAP measurement techniques and compare their repeatability, credibility, and clinical application to provide a reference for medical personnel to choose appropriate IAP measurement techniques.

## Materials and methods

### 2.1. Design

Given the underexplored nature of IAP measurement techniques and the broad nature of the review question, a scoping review was chosen as an appropriate approach for our study because it has the ideal scope to map the broad nature of the review issues in the research area [15]. This approach facilitates the collection of information from different sources and the design of studies involving various research issues. This scoping review was performed based on the five-step methodological framework proposed by Arksey and Malley [14]. We followed the Preferred Reporting Items for Systematic Reviews and Meta-Analysis extension for Scoping Reviews (PRISMA-ScR) checklist [16], which provides guidance on scope review reports and promotes study quality and rigor.

### 2.2. Stage 1: Identifying the research question

The methodological framework of the scoping review recommends the adoption of multiple research questions. The two specific questions we used to guide this review were 1) to identify and summarize the emerging direct and alternative methods for IAP measurement and 2) to assess the credibility and feasibility of these measurement methods.

### 2.3. Stage 2: Identifying the relevant studies

Electronic databases, including PubMed, EMBASE, Web of Science, EBSCO, Scopus and ProQuest, were systematically searched, only articles published from 2000 were considered, as this review is an update on the two previous reviews cited in the introduction section [6, 9]. A combination of Medical Subject Heading/Emtree terms and free terms was used. The search strategies of each database are provided in S1 File. Two independent reviewers followed the same strategies to perform the search process. The reference lists of the included articles were reviewed thoroughly to identify additional studies.

### 2.4. Stage 3: Determining study selection

The retrieved titles were imported into EndNote X9 for deduplication, and two trained researchers independently conducted preliminary screening by reading questions and screening abstracts based on the inclusion and exclusion criteria. Then, the entire text of each article was read for secondary screening. If there were any disagreements during the screening process, we discussed and resolved them with a third researcher and ultimately determined the articles that met the criteria. The study selection was based on the following inclusion and exclusion criteria.The inclusion criteria were 1) clinical studies of human subjects undergoing IAP measurement (including but not limited to adult and paediatric critically ill patients); 2) original studies focused on the validation (including but not limited to the repeatability, safety or agreement analysis) of IAP measurement techniques and (or) development; and 3) studies

published in English between 2000 and 2023. The exclusion criteria were 1) animal experiments or simple in vitro experiments; 2) studies conducted in newborns; 3) duplicate publications; and 4) conference abstracts, reviews, case reports, news, guidelines, letters, books or study protocols.

## 2.5. Stage 4: Charting the data

The research team first developed a standardized table (including first author, publication year and country, research design, research purpose, participant, sample size, measurement technique, and research result information) to extract the data. Pilot tests were then conducted on the included articles covering the two research questions until a consensus was reached. Next, the first author carefully read and evaluated each included article, while another reviewer repeatedly checked the article to ensure accuracy. The final extracted data were reviewed and discussed by the research team. We did not conduct a systematic quality assessment of the included studies because this was a scoping rather than a systematic review.

## 2.6. Stage 5: Collating, summarizing, and reporting the data

We used descriptive tables to synthesize and compare a wide range of findings to make it convenient for the authors to compare similarities and differences among the included studies. This iterative approach focused on the two specific objectives of this scoping review. A narrative approach was applied to the data synthesis. Research team members jointly participated in the discussion and revision of the findings and assisted in verifying the identification and description of themes throughout the process.

## Results

Electronic searches and a review of other sources yielded 9954 citations, resulting in 8668 unique citations for title and abstract screening following the removal of 1286 duplicates. Overall, 8540 publications were excluded based on title and abstract screening, which left 128 citations for full-text review. Ultimately, Forty-two studies were included in this review. A PRISMA flow diagram illustrating the search process is shown in S1 Fig.

### 3.1. Characteristics of the included studies

The main characteristics of each included study are provided in S1 Table, while the methods of some IAP measurement techniques are provided in S2 File. Forty-two articles concentrated on the development, feasibility and validity of IAP measurement techniques. These studies were conducted in the following countries: Canada (n = 4) [17–20], Germany (n = 8) [21–28], Israel (n = 3) [29–31], the USA (n = 7) [32–38], Belgium (n = 3) [39–41], the Netherlands (n = 3) [42–44], China (n = 2) [45, 46], Australia (n = 3) [47–49], Italy (n = 2) [50, 51], and Mexico (n = 2) [52, 53], with one study each from Singapore [54], Slovenia [55], New Zealand [56], France [57] and the United Arab Emirates [58]. The year of publication ranged from 2002 to 2023.

   **3.1.1. Study designs.**   All Forty-two included articles described cohort studies. Of these, thirty-eight were prospective studies [17–23, 26–40, 43–58], and four were retrospective studies [24, 25, 41, 42]. Two were multicentre research studies [19, 48], while the rest were single-centre [17, 18, 20–47, 49–58].

   **3.1.2. Study participants.**   Seventeen studies were conducted on adult ICU patients [19, 23, 34, 37, 40, 41, 44–52, 54, 57]; four studies were conducted on critically ill children (two assessed PICU patients [20, 23], one study assessed pediatric renal transplantation patients

[24], one assessed pediatric liver transplantation patients [25]); thirteen were conducted on patients undergoing abdominal surgery [18, 26–32, 38, 39, 42, 57, 58], while one was conducted on patients after abdominal surgery [50]; four were conducted on women [31, 33, 34, 51], and two were conducted on volunteers [17, 40]. The sample size in each study ranged from 5 to 1097 participants. The number of measurements in each study ranged from 25 to 6078. The included studies were mainly conducted in the university or hospital setting.

## 3.2. IAP measurement techniques

**3.2.1. Direct IAP measurement techniques.** Seven studies used a direct method for IAP measurement; of these, one used a handheld Stryker intracompartmental pressure monitor [18], two used a 14-Fr PVC round drain [32, 33], one used a single-lumen central venous catheter [44], one used a solid microsensor [51], one used an air-capsule technique [28] and one used piezoresistive pressure measurement [27].

**3.2.2. Indirect IAP measurement techniques.** A total of fourteen studies described intravesical methods for IAP measurement; of these, four studies used a commercial IAP monitoring system [24, 25, 32, 37], three studies used a pressure transducer [49, 52, 58], and seven studies used a fluid column measurement method [21, 26, 38, 41, 43, 53, 57]. Six studies described the intragastric pressure measurement method, of which one used air-capsule-based measurement [23], one used continuous intra-gastric monitoring system [22], one used a new nasogastric polyfunctional catheter [50], one used a new compliance catheter [39], one used an air-capsule technique [28] and one used a new device inserted between the nasogastric probe and enteral nutrition feeding pump and tubing [19]. One study described the intrarectal pressure measurement method [38], which involved a rectal T-DOC 7Fr air-filled balloon catheter connected to a computer displaying the IAP. Three studies described the femoral venous measurement method, of which two used a pressure transducer [20, 47] and one used a Foley manometer [48]. Four studies described the intravaginal measurement method, all of which involved wireless intravaginal pressure transducers [33, 35, 36, 56].

**3.2.3. Less invasive IAP measurement techniques.** Seven studies described less invasive IAP measurement techniques. Four described abdominal wall tension measurement techniques, of which one used transcutaneous sensors [55], one used a high-precision resistance strain pressure transducer [46], one used novel equipment involving a thrust metre and a homemade device to measure the required thrust to produce displacement [45], and one used a built-in force and distance sensor [42]. One study described microwave reflectometry [29], one described a less invasive ultrasonographic method [54], and one described a novel tool for less invasively characterizing pressurized, physiological vessels comprising a pressure sensor, a distance sensor, and luer-lock connections [17].

## 3.3. Clinical validation

Three studies were validated in vivo and in vitro [23, 42, 52], while only in vivo validation was conducted in the rest. Bland–Altman plots was used to conduct agreement analyses in twenty-five studies [17–23, 26–28, 32, 34, 37, 40, 43, 47, 48, 50–58], and safety validation was conducted in 6 studies [24, 25, 27, 28, 30, 41]. Four studies explored the time expenditure for IAP measurement [24, 25, 30, 41], and three explored the measure reproducibility, of which one evaluated the intraobserver and interobserver reproducibility [37], one evaluated the intraobserver reproducibility [42] and one evaluated the interobserver reproducibility [54]. The advantages, disadvantages and recommendations of each IAP measurement technique are described in S2 Table.

**3.3.1. Direct IAP measurement techniques.** Seven studies validated the direct method, all studies showed that the direct IAP measurements were feasible, fast, uncomplicated and no complications were found [18, 27, 28, 32, 33, 44, 51]. However, one study found that a hand-held Stryker intracompartmental pressure monitor connected to the peritoneal dialysis cathe-ter was not a reliable estimate of the insufflator pressures especially at lower pressures [18], small sample size and lack of a gold standard comparator may have affected this results.

**3.3.2. Indirect IAP measurement techniques.** Clinical validation showed that rectal IAP measurement has poor repeatability, and can not replace intravesical measurement method [38] (n = 1). Three studies validated the femoral venous measurement method (n = 3) [20, 47, 48], Bland-Altman analysis showed that the measurement agreement was extremely low between femoral venous pressure and intravesical pressure and cannot be recommended as a surrogate intravesical measurement method. Five studies validated the intragastric method (n = 5/6) using Bland-Altman analysis, among those, three studies showed that the intragastric method agreed favourably with intravesical method [19, 23, 28], while one study that measure-ments between intragastric method and direct measurement do not correlate well [22]. Three studies validated the intravaginal method without Bland-Altman analysis, one study was inter-nally validated and showed excellent repeatability across a range of activities [56], two studies were externally validated with rectal method and found the wireless intravaginal method was reliable and accurate comparing to rectal balloon catheters [35, 36]. For intravesical method, most studies(13/14) showed this technique was safe, reliable and accurate [21, 24–26, 32, 37, 38, 41, 43, 49, 52, 53, 57], while one study found that IAP measurements were less reliable while measurements for pressures higher than 12 mmHg [58].

**3.3.3. Less invasive IAP measurement techniques.** The abdominal wall tension measure-ment (AWT) method was validated in four studies, three studids found AWT has a good cor-relation with intra-vesical pressure or insufflator pressures [42, 45, 46]. However, one study showed the repeatibility of AWT measurement method was low with a mean CV of 14% for repeated measurements [42], while anotherstudy found it has low sensitivity and poor consis-tency with the gold standard method [55]. One study validated microwave reflectometry method, which was found low sensitivity [29]. The novel tool to non-invasively characterize pressurized, physiological vessels was proved to be reliable and valid in both cadavers and liv-ing participants, but the dynamic feasibility of the device requires further study [17]. The novel ultrasound-based IAP method was validated in one study, displaying good correlation and agreement between IAP and IBP at levels up to 15 mmHg [54].

## Discussion

Previous scholars have summarized IAP measurement techniques prior to 2000 [6, 9], so this study only includes articles from 2000 onwards. The importance of this review is appar-ent given that relevant guidelines and consensus emphasize the importance of IAP measure-ment in IAH/ACS research and management [2, 10]. IAP measurement methods can be divided into three categories: direct, indirect and less invasive The bladder, stomach, femo-ral vein, vagina and rectum have been described as routes for indirect measurement. How-ever, the number of clinical studies that validate or evaluate aspects of IAP measurement is low.

Direct measurement method was considered to be the most accurate method [8]. The use of abdominal drains for postoperative direct IAP measurement has been reported by Risin et al. [31], in which a 14-Fr PVC round drain was connected to the invasive blood pressure measurement system under sterile conditions. The obtained IAP values correlated well with the pressure obtained from the insufflator, and no complications were observed.

Unfortunately, no Bland-Altman analysis was performed. In another study, the same authors reported good correlation between the directly measured IAP and pressure measured through the transvesical method in patients undergoing laparoscopic procedures [30]. Recently, Thangarasa et al. [18] reported a handheld Stryker intracompartmental pressure monitor connected to the peritoneal dialysis catheter to directly measure IAP in peritoneal dialysis patients. As this method cannot reliably estimate insufflator pressures, especially at lower pressures, further studies are required to determine an ideal IAP measurement tool to guide peritoneal dialysis management. Although direct method was proved to be accurate, fast and simple, the cost and invasiveness should be considered in clinical practice, validation of this technique should also be performed in postoperative patients. Therefore, it is not recommended as a routine method for IAP measurement and should be prioritized for critically ill patients where standard techniques are contraindicated or could be inaccurate.

Traditionally the bladder has been used as the method of choice for IAP measurement.

The technique was proposed by Kron in 1984 and originally described as an open bedside single IAP measurement system [6], which means a lot of time-consuming manipulations that disrupt a closed sterile system at every measurement and increase potential risk of urinary tract infection or sepsis. In 1998, Cheatham reported a revision of Kron's original technique. This closed system repeated measurement technique is safer, less invasive, more efficient and more cost-effective [59]. Although an IAP can be obtained at regular intervals, this technique is ideal for screening and monitoring for a short period of time (a couple of days) because of leakage. Additionally, it remains labour intensive, especially when hourly IAP measurements are needed. Researchers are beginning to recognize the value of continuous IAP monitoring. In 2004, Balogh et al. [49] introduced a method for continuous IAP measurement using a size 18-Fr standard three-way Foley catheter, which was found to perform excellently in ICU patients. The issues of urine drainage and correct IAP measurement have not been resolved since the level of the urine collection bag 30 cm below the symphysis pubis will create a negative "suction" pressure, resulting in underestimation of the true IAP [6]. Furthermore, most ICUs found it unreasonable to replace the existing Foley catheter with this larger and more expensive 18-Fr catheter. In most cases, intermittent measurement may be sufficient. However, when patients with ACS need emergency abdominal decompression, continuous measurement may be preferable. Therefore, continuous measurement is not suited for screening IAH, but is best for long-term continuous fully automated monitoring IAP, as it is less prone to errors and most cost-effective if in place for a longer period of time.

The transgastric method can be used when patients have no Foley catheters in place or when accurate bladder pressure is not possible due to insufficient bladder compliance, including neurogenic bladder, bladder trauma, bladder tumor. Collee et al. [60] used a fluid column in the nasogastric tube to measure IAP, the advantages are that it is cheap, no interference with urine output, and no risks of infection. The biggest potential issue of this technique is that it may affect nasal feeding, and theoretical complications with traditional nasogastric tubes, such as pneumonia, oesophageal or gastric perforation and malposition with aspiration. Subsequently, this technique has been replaced by the use of a balloon-tipped stomach catheter with the Spiegelberg, ACM-IGP (Spiegelberg company, Germany) or CiMON device (Pulsion Medical Systems, Munich, Germany–www. pulsion.com). Otto et al. [28] reported an air-capsule technique for the direct measurement of IAP in adult ICU patients. IAP values based on air-capsule-based measurement agreed well with the transgastric measurement technique in elevated IAP up to 17 mmHg. Kaussen and colleagues [23] combined ACM with IGP determination in critically ill children for the first time. In vitro and in vivo validation indicated that ACM-IGP may be even more suitable for measuring IAP, especially in high-risk clinical

settings for IAH. This technique is simple, accurate, fast, repeatable, fully automated and being free from interference caused by wrong transducer positions, but the validation in humans is still in its children stage.

Increasing evidence has been obtained from adult medicine studies in recent years indicating that the measurement accuracy, reliability and sensitivity of the transrectal and transfemoral vein methods remain insufficient and cannot be recommended as surrogate IAP measures. From a theoretical perspective, using rectal catheters for IAP measurement appears to be less invasive and potentially useful in ambulatory settings and pregnant patients. A validation study conducted by Staelens et al. [40] on the transrectal IAP measurement technique in ICU patients showed that IAP estimation using a rectal catheter was unfeasible because the IAP values cannot be trusted or validated, and the failure rate of obtaining reproducible IAP values via transrectal measurement is very high. There are conflicting data regarding whether the transfemoral vein measurement technique is an appropriate alternative method for IAP measurements. In an animal model, Regli et al. [61] demonstrated a good correlation between IAP and femoral venous pressure, but it seems difficult to transfer this approach to the human population. Howard et al. [47] found that the transfemoral vein measurement technique cannot be recommended as a surrogate measure for IAP measurement via the standard intravesical technique even at IAP >20 mmHg in adult ICU patients. A recent study conducted by Gutting et al. [20] also demonstrated the unreliability of transfemoral vein measurement in critically ill children. Wireless transvaginal measurements have also been studied. This technique overcomes limitations of traditional urodynamic testing, whose wireless functions, shape and ease of use facilitate the quantitative IAP measurement during exercise and activity, which laying the foundations for monitoring IAP outside of the clinic environment [36]. However, the accuracy of this technique in measuring absolute IAP cannot be determined as there is not a clinical gold standard for IAP measurement as a reference. So this technique is mainly applicable to exploring the role of elevated IAP in the progression, recurrence, and incidence of pelvic floor disorder during exercise and daily activities, but is not suitable for routine monitoring of IAP.

The AWT method has yielded the most data in previous studies. The AWT is affected by the contents of the abdominal cavity. When the contents of the abdominal cavity increase or abdominal infection affects the peritoneum, the AWT increases [38, 62]. The correlation between abdominal wall tension and IAP has long been studied. In a preliminary experiment conducted by van Ramshorst et al. [63], the AWT of 7 points on the abdominal wall of 2 cadavers was measured for the first time using noninvasive AWT measurement equipment with a tensiometer. They found a significant correlation between AWT and IAP, especially in the middle abdomen. The AWT method is very fast, accurate, simple and noninvasive, which could be performed by nurses to screen for IAH without the need for a long period of training or supervision. However, AWT measurement devices should be standardized to obtain more reliable and repeatable evaluations. Future studies should also clarify the effects of common factors, such as gender, BMI and muscle relaxants and mechanical ventilation on AWT in the ICU. The microwave reflection method is considered one of the most promising methods for IAP measurement [64]. Recently, David et al. [29] developed a microwave reflectometry system, and a proof-of-concept clinical trial was conducted on five patients during laparoscopic surgery. The system could assess changes in IAP; however, the measurement of absolute IAP values remained a challenge. Further research is necessary to optimize the sensitivity of this system. Less invasive technology will become a new trend in the development of IAP measurement technology in the future and has substantial potential to replace traditional IAP measurement technologies.

### 4.1. Limitations

This review has several limitations. Although we searched six databases, including grey literature, which can be an important information source, and thoroughly reviewed the reference lists of the identified studies, we only included studies published from 2000 as this review is an update on the two previous reviews cited in the introduction section [6, 9]. Therefore, the information provided in this article possibly affecting a systematic understanding of IAP measurement techniques. In addition, although some studies (3/42) involved the clinical validation of IAP measurement techniques, specific experimental data on consistency, feasibility and safety could not be extracted; however, our study emphasized obtaining a broad overview of current IAP measurement techniques. Therefore, we included research on the validation of all IAP measurement techniques. Finally, no research quality evaluation was conducted regarding the characteristics and methods of scoping reviews [14]. However, the purpose of a scoping review is to provide a general map of the literature on general topics to inform readers and clarify the need for further research [65], which has been achieved here.

## Conclusion

Overall, forty-two studies were included in this scoping review. We provided a comprehensive and broad overview of the current and emerging direct and alternative methods for IAP measurement. However, clinical data regarding the validation of new IAP measurement methods or the reliability of established measurement techniques remain lacking. Direct measurement is not recommended as a routine method for IAP measurement considering the cost and invasiveness. The transrectal and transfemoral vein methods cannot be recommended as surrogate IAP measures due to the insufficient accuracy, reliability and sensitivity. The intravesical method, which has been studied most extensively, can be used as reliable route for continuous and intermittent IAP measurement. Less invasive technology will become a new trend to measure IAP in the future and has substantial potential to replace traditional IAP measurement technologies, but further validation and standardization are still needed. For example, a commercially available AWT measurement device may become a powerful tool for IAP measurement in the future. Health care professionals should choose appropriate measurement tools based on the advantages and disadvantages of each IAP measurement technique in combination with assessing specific clinical situations.

## Supporting information

**S1 File. Search strategies of each database.**
(DOCX)

**S2 File. Summary of measurement methods.**
(DOCX)

**S1 Table. Characteristics of included studies.**
(DOCX)

**S2 Table. The advantages, disadvantages and recommendations of each IAP measurement technique.**
(DOCX)

**S1 Fig. PRISMA flow diagram for included studies selection.**
(PDF)

## Author Contributions

**Conceptualization:** ZhiRu Li.

**Methodology:** ZhiRu Li.

**Supervision:** HuaFen Wang, FangYan Lu.

**Writing – original draft:** ZhiRu Li.

**Writing – review & editing:** HuaFen Wang, FangYan Lu.

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
