## [Decision Letter · Decision Letter 0]

19 Nov 2023

PONE-D-23-34086The development, feasibility and credibility of intra-abdominal pressure measurement techniques: A scoping reviewPLOS ONE

Dear Dr. Wang,

Thank you for submitting your manuscript to PLOS ONE. After careful consideration, we feel that it has merit but does not fully meet PLOS ONE’s publication criteria as it currently stands. Therefore, we invite you to submit a revised version of the manuscript that addresses the points raised during the review process.

We look forward to receiving your revised manuscript.

Kind regards,

Fei Yan

Academic Editor

PLOS ONE

Journal Requirements:

Reviewers' comments:

Reviewer's Responses to Questions

**Comments to the Author**

1. Is the manuscript technically sound, and do the data support the conclusions?

Reviewer #1: Yes

Reviewer #2: Yes

2. Has the statistical analysis been performed appropriately and rigorously? 

Reviewer #1: N/A

Reviewer #2: Yes

3. Have the authors made all data underlying the findings in their manuscript fully available?

Reviewer #1: Yes

Reviewer #2: Yes

4. Is the manuscript presented in an intelligible fashion and written in standard English?

Reviewer #1: Yes

Reviewer #2: Yes

5. Review Comments to the Author

Reviewer #1: This review article on the "Development, Feasibility, and Credibility of Intra-Abdominal Pressure Measurement Techniques" is a comprehensive review that covers a wide range of direct and alternative measurement methods. The article is rigorous in its search and filtration of relevant literature, analyzing the feasibility, safety, and effectiveness of various measurement techniques. However, certain areas require further elaboration and clarification. Here are my suggestions for improvements.

1.In the Abstract and Conclusion, the viewpoints are unclear: Although some disadvantages of the existing IAP measurement techniques are pointed out in the conclusion, it does not clearly propose directions for improvement or new research suggestions. Also, the view that the "Transvesical method is the potentially most credible and easily applicable technique" should be made clearer, for example, through comparison or data support.

2.The Introduction contains redundant information and lacks clear organization: While the authors have detailed the importance and impact of IAP, the discussion on the reliability, safety, and accuracy of IAP measurement methods between lines 78-94 appears repetitive and poorly organized. For instance, the descriptions in lines 79-82 and 87-94 are redundant. The authors need to organize the information more effectively to avoid repetition and ensure each paragraph has a clear theme and objective.

3.The Results section lacks conclusions or interpretations: The Results section should include interpretations of the data or meaning of the results, not just descriptions of the results. For instance, in the clinical validation part (lines 229-236), the authors list various validation methods but do not provide interpretations or conclusions of these validation results.

4.In the Discussion section, the pros and cons of various IAP measurement methods need more detailed elaboration. For example, in lines 311-314, the authors mention the vaginal measurement method but do not discuss its limitations thoroughly, such as it being limited to female populations, and the intrarater reliability has not yet been evaluated.

5.Although the study includes research from 2000 to 2023, limiting it to this period may overlook some early key studies, possibly affecting a comprehensive understanding of IAP measurement techniques (lines 336-339). Please explain why only the studies from 2000 to 2023 were chosen and try to add related content in the limitations section.

6.There are several grammatical issues in the paper. For example, the phrase in lines 60-61, "Abdominal wall tension method still need be standardized" should be changed to "Abdominal wall tension method still needs to be standardized". And in lines 59-60, the sentence "transrectal and transfemoral vein methods have not be proven to be feasible and credible." should be corrected to "have not been proven to be feasible and credible."

Reviewer #2: This manuscript conducted a very comprehensive and promoting review of intra-abdominal pressure measurement, with three major categories of method including direct, indirect and less invasive. We recommand the following revisions:

1. The conclusion section "should choose appropriate measurement tools based on the advantages and disadvantages of each IAP measurement technique in combination with assessing specific clinical situations" is still weak and lack of direct reference value. We suggest to make conclusion more in details, in order to provide direct and brief assistance for health care professionals.

2. The included studies were listed very in details in supplementary materials, but too long. I suggest the addition of a shorter table summarize the major advantages and disadvantages, accuracy and other characteristics of the three catogories of method.

3. "IAP should be measured in a standardized and repeatable manner". Sure, but after this very comprehensive review, I'm looking forward some specific developing trend advice from authors. This would help improvement of current studies.

6. PLOS authors have the option to publish the peer review history of their article (what does this mean?). If published, this will include your full peer review and any attached files.

Reviewer #1: No

Reviewer #2: No

---

## [Author Response · Author response to Decision Letter 0]

27 Dec 2023

Comment 1: In the Abstract and Conclusion, the viewpoints are unclear: Although some disadvantages of the existing IAP measurement techniques are pointed out in the conclusion, it does not clearly propose directions for improvement or new research suggestions. Also, the view that the "Transvesical method is the potentially most credible and easily applicable technique" should be made clearer, for example, through comparison or data support.

Response: We are grateful to the reviewers' suggestion. In the Abstract and Conclusion, we have added directions for improvement or new research suggestions. And through comparing the major advantages and disadvantages, accuracy and other characteristics of the three catogories of method, we have come to this conclusion that 

"Transvesical method is the potentially most credible and easily applicable technique".

Comment 2: The Introduction contains redundant information and lacks clear organization: While the authors have detailed the importance and impact of IAP, the discussion on the reliability, safety, and accuracy of IAP measurement methods between lines 78-94 appears repetitive and poorly organized. For instance, the descriptions in lines 79-82 and 87-94 are redundant. The authors need to organize the information more effectively to avoid repetition and ensure each paragraph has a clear theme and objective.

Response: Thanks for the suggestion from the reviewers. We have reorganized the introduction section to ensure that each paragraph has a clear theme and goal.

Comment 3: The Results section lacks conclusions or interpretations: The Results section should include interpretations of the data or meaning of the results, not just descriptions of the results. For instance, in the clinical validation part (lines 229-236), the authors list various validation methods but do not provide interpretations or conclusions of these validation results. 

Response: We are grateful to the reviewers' comment. In the clinical validation part, We have added detailed explanations on the validation results of direct IAP measurement techniques, indirect IAP measurement techniques and less invasive IAP measurement techniques.

Comment 4: In the Discussion section, the pros and cons of various IAP measurement methods need more detailed elaboration. For example, in lines 311-314, the authors mention the vaginal measurement method but do not discuss its limitations thoroughly, such as it being limited to female populations, and the intrarater reliability has not yet been evaluated.

Response: We sincerely appreciate the reviewers' suggestion. In the discussion section, we have provided a more detailed explanation of the advantages, disadvantages and applicable clinical situations of each IAP measurement method, hoping to provide reference for clinical staff to choose appropriate IAP measurement tools. 

Comment 5: Although the study includes research from 2000 to 2023, limiting it to this period may overlook some early key studies, possibly affecting a comprehensive understanding of IAP measurement techniques (lines 336-339). Please explain why only the studies from 2000 to 2023 were chosen and try to add related content in the limitations section.

Response: We are grateful to the reviewers' comment. In this review, we only considered studies published from 2000 as we have extended the data from two previous reviews cited in the introduction section. Nonetheless, the information provided in this article possibly affecting a systematic understanding of IAP measurement techniques. And we have added explaination in the limitations section.

Comment 6: There are several grammatical issues in the paper. For example, the phrase in lines 60-61, "Abdominal wall tension method still need be standardized" should be changed to "Abdominal wall tension method still needs to be standardized". And in lines 59-60, the sentence "transrectal and transfemoral vein methods have not be proven to be feasible and credible." should be corrected to "have not been proven to be feasible and credible."

Response: We appreciate the kind comment. We have modified the grammatical issues in the paper.

Comment 7: The conclusion section "should choose appropriate measurement tools based on the advantages and disadvantages of each IAP measurement technique in combination with assessing specific clinical situations" is still weak and lack of direct reference value. We suggest to make conclusion more in details, in order to provide direct and brief assistance for health care professionals.

Response: We are grateful to the reviewers' comment. We have added detailed 

conclusions on the advantages, disadvantages and applicable clinical situations of 

each IAP measurement method in the discussion and conclusion section. Also, we 

have added two shorter tables in the attachment, summarizing the major advantages 

and disadvantages, recommendations and other characteristics of the three catogories 

of method

Comment 8：The included studies were listed very in details in supplementary materials, but too long. I suggest the addition of a shorter table summarize the major advantages and disadvantages, accuracy and other characteristics of the three catogories of method.

Response: We sincerely appreciate the reviewers' suggestion. We have added two shorter tables in the attachment, summarizing the major advantages and disadvantages, recommendations and other characteristics of the three catogories of method.

Comment 9："IAP should be measured in a standardized and repeatable manner". Sure, but after this very comprehensive review, I'm looking forward some specific developing trend advice from authors. This would help improvement of current studies.

Response: We sincerely appreciate the reviewers' suggestion. The view that “IAP should be measured in a standardized and repeatable manner” is difficult to elaborate in detail in this review, as the accuracy of different measurement methods is affected by different factors. Therefore, we only briefly gave an example of standardized measurement of abdominal wall tension in the discussion section.

---

## [Decision Letter · Decision Letter 1]

16 Jan 2024

The development, feasibility and credibility of intra-abdominal pressure measurement techniques: A scoping review

PONE-D-23-34086R1

Dear Dr. Wang,

We’re pleased to inform you that your manuscript has been judged scientifically suitable for publication and will be formally accepted for publication once it meets all outstanding technical requirements.

Kind regards,

Fei Yan

Academic Editor

PLOS ONE

Additional Editor Comments (optional):

Reviewers' comments:

Reviewer's Responses to Questions

**Comments to the Author**

1. If the authors have adequately addressed your comments raised in a previous round of review and you feel that this manuscript is now acceptable for publication, you may indicate that here to bypass the “Comments to the Author” section, enter your conflict of interest statement in the “Confidential to Editor” section, and submit your "Accept" recommendation.

Reviewer #1: All comments have been addressed

Reviewer #2: All comments have been addressed

2. Is the manuscript technically sound, and do the data support the conclusions?

Reviewer #1: Yes

Reviewer #2: Yes

3. Has the statistical analysis been performed appropriately and rigorously? 

Reviewer #1: N/A

Reviewer #2: Yes

4. Have the authors made all data underlying the findings in their manuscript fully available?

Reviewer #1: Yes

Reviewer #2: (No Response)

5. Is the manuscript presented in an intelligible fashion and written in standard English?

Reviewer #1: Yes

Reviewer #2: Yes

6. Review Comments to the Author

Reviewer #1: Thank you for you in addressing the comments. your revisions demonstrated a careful consideration of the suggestions provided. The changes have positively impacted the overall clarity and coherence of the paper.

Reviewer #2: (No Response)

7. PLOS authors have the option to publish the peer review history of their article (what does this mean?). If published, this will include your full peer review and any attached files.

Reviewer #1: No

Reviewer #2: No

---

## [Editor Report · Acceptance letter]

12 Mar 2024

PONE-D-23-34086R1 

PLOS ONE

Dear Dr. Wang, 

I'm pleased to inform you that your manuscript has been deemed suitable for publication in PLOS ONE. Congratulations! Your manuscript is now being handed over to our production team.

Kind regards, 

on behalf of

Dr. Fei Yan 

Academic Editor

PLOS ONE